# A Novel Predictive Tool for Discriminating Endometriosis Associated Ovarian Cancer from Ovarian Endometrioma: The R2 Predictive Index

**DOI:** 10.3390/cancers13153829

**Published:** 2021-07-29

**Authors:** Naoki Kawahara, Ryuta Miyake, Shoichiro Yamanaka, Hiroshi Kobayashi

**Affiliations:** Department of Obstetrics and Gynecology, Nara Medical University, Nara 634-8521, Japan; ryuta-miyake@naramed-u.ac.jp (R.M.); shoichiroyamanaka@naramed-u.ac.jp (S.Y.); hirokoba@naramed-u.ac.jp (H.K.)

**Keywords:** ovarian endometrioma, endometriosis associated ovarian cancer, magnetic resonance imaging, MR relaxometry, CEA, R2 predictive index

## Abstract

**Simple Summary:**

Ovarian Endometrioma (OE) is a precancerous condition for endometriosis-associated ovarian cancer (EAOC). For many clinicians observing OE outpatients, setting the appropriate time for surgery can be a challenge because there is no suggestive milestone. Out of the fear of malignant transformation, many patients have surgery conducted according to respective faculty standards. This study aims to investigate a novel, noninvasive method not requiring an MRI device. This study partly helps to lift the above restrictions, and has the potential to suggest intervention-appropriate timing to the physician.

**Abstract:**

Background: Magnetic resonance (MR) relaxometry provides a noninvasive tool to discriminate between ovarian endometrioma (OE) and endometriosis-associated ovarian cancer (EAOC), with a sensitivity and specificity of 86% and 94%, respectively. MRI models that can measure R2 values are limited, and the R2 values differ between MRI models. This study aims to extract the factors contributing to the R2 value, and to make a formula for estimating the R2 values, and to assess whether the R2 predictive index calculated by the formula could discriminate EAOC from OE. Methods: This retrospective study was conducted at our institution from November 2012 to February 2019. A total of 247 patients were included in this study. Patients with benign ovarian tumors mainly received laparoscopic surgery, and the patients suspected of having malignant tumors underwent laparotomy. Information from a chart review of the patients’ medical records was collected. Results: In the investigative cohort, among potential factors correlated with the R2 value, multiple regression analyses revealed that tumor diameter and CEA could predict the R2 value. In the validation cohort, multivariate analysis confirmed that age, CRP, and the R2 predictive index were the independent factors. Conclusions: The R2 predictive index is useful and valuable to the detection of the malignant transformation of endometrioma.

## 1. Introduction

Endometriosis is defined as the presence of endometrial glands and stroma outside of the uterus, and it is most often detected in the pelvic peritoneum and ovaries. Repeated hemorrhages in the peritoneum or ovaries may contribute to not only the symptoms of dysmenorrhea, chronic pelvic pain, and infertility, which negatively affect the patients’ quality of life [1], but also to the subsequent risk of developing endometriosis-associated ovarian cancer (EAOC) [2,3,4]. There is also evidence of an epidemiologic link between iron overload and the various types of human carcinoma, including malignant mesothelioma, renal cell carcinoma, hepatocellular carcinoma, and EAOC [5]. The local accumulation of hemoglobin, heme, and iron species causes severe oxidative stress and antioxidant depletion, leading to distortion in the redox balance [6,7,8,9]. Previous studies have shown that the patients with EAOC had lower levels of Hb and iron-related compounds than those with benign ovarian endometrioma (OE) [10,11]. The total iron levels of cyst fluid can discriminate EAOC from OE with a cutoff point of 64.8 mg/L (sensitivity, 85%; specificity, 98%) [12]. Furthermore, liver and heart iron content were reliably quantified by the measurement of the transverse magnetic relaxation rate R2 or R2* value, using complex, chemical shift-encoded MR examination [13]. MR relaxometry could be a noninvasive, preoperative prediction tool, and showed a favorable predictive accuracy for the malignant transformation with a cutoff point of 12.1 (sensitivity, 86%; specificity, 94%) [12,14]. However, MR relaxometry could also be a major challenge because of the inconsistency of the cut-off R2 value, which depends on the device manufacturer, the inability to set this device in smaller facilities, and the impracticality of repeated imaging. This study aims to extract the factors contributing to the R2 value, to make a formula for estimating the R2 value, and to assess whether the R2 predictive index calculated by the formula could discriminate EAOC from OE without MR relaxometry.

## 2. Materials and Methods

### 2.1. Patients

A list of patients with primary, previously untreated, histologically-confirmed ovarian tumors who were treated at Nara Medical University Hospital between November 2012 and February 2019 was generated from our institutional registry. The consent form for patients’ data availability for research use was obtained at the first hospitalization, and after approval by the Ethics Review Committee of the Nara Medical Hospital, and the opt-out form was provided through our institutional homepage. The consent form for the case report was obtained from the patient as well. The current study consisted of two cohorts: the investigative and validation cohorts. The investigative cohort included 142 patients with newly diagnosed ovarian tumors, who had undergone a preoperative MR relaxometry before surgery to assess the correlation between the R2 value and the following factors, and to make a formula predicting the R2 value. In the validation cohort, a total of 105 patients without MR relaxometry were included to confirm the usefulness of the obtained formula in discriminating EAOC from OE. No patients underwent surgery, chemotherapy, or radiotherapy for the ovarian tumors prior to treatment. Patients with benign ovarian tumors mainly received laparoscopic surgery, and the patients suspected of having malignant tumors underwent laparotomy. The following factors were collected through a chart review of the patients’ medical records: age, body mass index, parity, postoperative diagnosis (including FIGO stage), the date of surgery, tumor diameter, menopausal status, and pre-treatment blood test results including carbohydrate antigen 125 (CA 125), carbohydrate antigen 19-9 (CA 19-9), cancer antigen 72-4 (CA72-4), carcinoembryonic antigen (CEA), alpha-fetoprotein (AFP), and squamous cell carcinoma antigen (SCC) as tumor markers.

### 2.2. MR Relaxometry for Determining the R2 Value

All patients underwent routine MR imaging using T1W and T2W sequences. MR images were obtained on a 3T system (Magnetom Verio or Skyla, Siemens Healthcare, Erlangen, Germany). After the routine clinical MR imaging, the registered patients underwent MR relaxometry using the single-voxel acquisition mode sequence at multiple echo times, and by fitting an exponential decay to the echo amplitude at different multiple echo times [13]. A parameter R2 value (s-1) was calculated using high-speed T2 *-corrected multi-echo MR sequence (HISTO) by the 3T-MR system in vivo, which has been described previously [15,16]. The HISTO sequence was based on the single voxel steam sequences that could be used for relative fat quantification in the liver [17]. This sequence allows estimation of liver iron deposition since the T2 of water changes with iron concentration. The pulse sequence design and programming were done with an imaging platform (Siemens Medical Systems, Erlangen, Germany) and applied to the 3T-system. The sequence has a fixed number of five measurements with different TEs, which are as follows: 12, 24, 36, 48, and 72 ms. The typical protocol is performed in breath-hold, with a total acquisition time of 15 s. The repetition time (TR) was fixed to 3000 ms, which proved to be enough to compensate for the effects of the signal saturation while maintaining an acceptable acquisition time. A 15 × 15 × 15-mm spectroscopy voxel (VOI) was placed to select a region encompassing the liquid portion, but not the solid portion, of the cyst lumen. The study does not include patients with cystic lesions smaller than 15 mm. The largest cyst fluid was measured if there were any patients who had more than one cyst. The VOI could be located in the center of the OE or EAOC cyst by a radiologist who specializes in female pelvic MR imaging.

### 2.3. Statistical Analysis

The analyses were performed by SPSS version 25.0 (IBM SPSS, Armonk, NY, USA). In the first cohort, the correlation between the R2 value and several factors was assessed by regression analysis. Among the factors showing a correlation, a regression analysis was conducted to make a formula estimating the R2 value. In the second cohort, the receiver operating characteristic (ROC) curve analysis was performed to determine the cut-off value for predicting EAOC. The cut-off value was based on the highest Youden index (i.e., sensitivity + specificity − 1). We next used logistic regression analysis and a Cox proportional hazards model to assess the risk factors for EAOC. A two-sided *p* < 0.05 was considered to indicate a statistically significant difference.

## 3. Results

### 3.1. Patients

From November 2012 to February 2019, a total of 247 patients included in this study were divided as follows: 142 with MR relaxometry as the first cohort and 105 patients without MR relaxometry as the second cohort. The demographic and clinical characteristics of the investigative and validation cohort are outlined in Table 1 and Table 2, respectively. In the first cohort, the post-operative diagnosis of OE was found in 95 (66.9%) and EAOC in 32 (22.5%) patients. In this cohort, MR relaxometry discriminated EAOC from OE with a sensitivity and specificity of 85.3% and 75.0%, respectively. In the second cohort, the post-operative diagnosis of OE was found in 54 (51.4%) and EAOC in 51 (48.6%) patients. In either cohort, there was significant differentiation in age, cyst size, and menopausal status among benign and malignant cases, or OE and EAOC cases (*p* < 0.001). Furthermore, in the second cohort, non-parous was more common in OE patients (*p* = 0.031).

### 3.2. Correlation between the R2 Value and Pe-Surgical Factors

The regression analysis was conducted to make a model formula for predicting the R2 value from each pre-surgical factor. The R2 values were related to the following factors: [R2 value] = 33.23 − 3.37×10^−1^ × [Age (year)], r = 0.315, *p* < 0.001; [R2 value] = 25.80 − 7.90×10^−2^ × [Tumor diameter (mm)], r = 0.278, *p* = 0.001; [R2 value] = 21.23 − 1.70 × [CEA (ng/mL)], r = 0.262, *p* = 0.006; [R2 value] = 20.49 − 5.00 × 10^−3^ × [CA125 (U/mL)], r = 0.201, *p* = 0.020; [R2 value] = 12.85 + 4.00 × 10^−3^ × [Lymphocyte (/µL)], r = 0.186, *p* = 0.036; [R2 value] = −0.67 + 1.66 × [Hemoglobin (g/dL)], r = 0.185, *p* = 0.031; [R2 value] = 29.77 − 3.65 × 10^−1^ × [Platelet (×104/µL)], r = 0.197, *p* = 0.022; [R2 value] = −15.72 + 8.00 × [Albumin (g/dL)], r = 0.198, *p* = 0.024; and [R2 value] = 21.50 − 1.75 × [D-dimer (µg/mL)], r = 0.262, *p* = 0.003. The multiple regression analyses revealed that the tumor diameter and CEA could predict the R2 value as follows (Equation (1)):
[R2 predictive index] = 27.27 − 7.90×10^−2^ × [Tumor diameter (mm)] − 1.31 × [CEA (ng/mL)]
(1)

### 3.3. The Usefulness of R2 Predictive Index in Discriminating OE and EAOC

We next assessed the compatibility in discriminating OE and EAOC in the second cohort. Using the ROC curve, the cut-off values discriminating EAOC from OE are shown in Table 3. The cut-off value from the above formula was 18.70 (sensitivity, 83.2%; specificity, 76.4%; AUC = 0.816, *p* < 0.001) (Figure 1). Some factors indicating EAOC were extracted by univariate analysis (Table 3). A multivariate analysis confirmed that age, CEA, and CRP were extracted as independent factors for predicting malignant tumors (hazard ratio (HR): 22.15, 95% confidence interval (CI): 5.02–97.6, *p* < 0.001; HR: 4.49, 95% CI: 1.26–15.99, *p* = 0.020; HR: 11.18, 95% CI: 2.80–44.64, *p* = 0.001, respectively). Furthermore, including the R2 predictive index instead of tumor size and CEA, a multivariate analysis showed that age, CRP, and the R2 predictive index were the independent factors (HR: 17.20, 95% CI: 3.84–77.16, *p* < 0.001; HR: 6.76, 95% CI: 1.58–28.89, *p* = 0.010; HR: 8.25, 95% CI: 2.13–32.02, *p* = 0.002, respectively) (Table 4).

## 4. Case Report

A 31-year-old primigravida Japanese female was referred to our outpatient clinic with left side ovarian endometrioma (47 × 31 mm). She had no history of gynecological diseases such as endometriosis, abdominal pain, or abdominal surgery. On physical exam, external genitalia, vagina, and cervix were unremarkable; however, a pelvic exam revealed a non-tender mass in the left adnexal region. Figure 2a shows a characteristic sonographic finding with a clinically suspicious ovarian tumor. The CA125, CA19-9, and CEA assays were 44 U/mL, 24 U/mL, and 0.9 ng/mL, respectively. The R2 predictive index (22.38) suggested benign ovarian endometrioma. She presented to our hospital for every 3-month follow-up of the left endometrioma. Two years after the first administration, ultrasonography showed an enlarged ovarian cystic tumor (96 × 79 mm) with a solid portion (Figure 2b), which was indistinguishable from a solid part or clot. The levels of CA125, CA19-9, and CEA were 30 U/mL, 14 U/mL, and 1.1 ng/mL, respectively, and the R2 predictive index (18.25) suggested a malignant transformation. An MRI identified the solid tissue components, detected with enhancing on dynamic contrast-enhanced images (Figure 2c). The patient was evaluated by a gynecologic oncologist and radiologist. The patient was advised to have an MR relaxometry of the pelvis. After obtaining informed consent for conducting an MR relaxometry, an R2 value showed as 6.61 s-1 in the cyst (Figure 2d), which suggested a malignant tumor rather than benign ovarian endometrioma. She finally underwent surgical treatment including hysterectomy, adnexectomy, omentectomy, retroperitoneal, and para-aortic lymphadenectomy. The final histopathological results revealed endometrioid carcinoma within an endometrioma, International Federation of Gynecology and Obstetrics stage IA. Peritoneal washings were negative for malignant cells.

## 5. Discussion

We previously reported that MR relaxometry could be a noninvasive preoperative prediction tool and showed a favorable predictive accuracy for malignant transformations, with a sensitivity and specificity of 86% and 94%, respectively [12,14]. However, the differentiation of EAOC from OE using MR relaxometry could have major difficulty because of the inconsistency of the cut-off R2 value, which depends on the devise manufacture, the inability to set this device in smaller facilities, and the impracticality of repeated imaging. This study partly helps to overcome the above restrictions, and has the possibility to suggest appropriate intervention timing to physicians.

Firstly, tumor diameters showed a negative correlation with the R2 value in the first cohort. To refer to previous studies, the R2 value did not correlate with tumor diameters [10]. Similar to the method of a previous study dividing the cohort into OE and EAOC groups, we re-analyzed the first cohort by each group, and as a result, confirmed no correlation existed between the R2 value and tumor diameters.

Secondary, the R2 predictive formula includes CEA instead of CA125. CA125 is an important indicator of ovarian cancer that plays an irreplaceable role. CA125 levels increase in advanced-stage disease because of an increase in tumor burden [18], and it is useful in predicting chemotherapy responses, disease progression, and recurrence [19,20]. High levels of CA-125 can predict advanced-stage disease or suboptimal debulking. However, CA-125 levels are relatively low in early-stage OCCC in comparison with other histologic types of epithelial ovarian cancer because of the apparent initial smaller volume of disease, as well as the fundamental differences in the biology of malignancy [21,22,23]. For instance, Udomsinkul et al. reported in the case-control study that univariate analysis showed that higher CA125 levels were a significant factor for discriminating EAOC from a benign tumor, but it did not remain so in multivariate analysis [24]. One of the reasons for this result is that benign OE cases show a high CA125 level. Our study similarly showed that CA125 neither correlated with the R2 value in the first cohort nor gave cut-off values in the second cohort. On the other hand, CEA is a broad-spectrum tumor marker, and an increasing number of studies have suggested that CEA is strongly related to the diagnosis and prognosis of malignant tumors. CEA is reported as an independent predictor for identifying epithelial ovarian cancer and ovarian metastases [25]. Further studies found that the cut-off value of CEA in the differential diagnosis of primary ovarian tumors and metastatic ovarian cancer was 2.33 μg/L [26]. Among EAOC tumors, CCC shows a lower CEA level than that of endometrioid carcinoma, and the cut-off of 3.270 μg/L can discriminate endometrioid carcinoma from CCC [27]. Our study found that in the second cohort, the CEA level of EAOC was significantly higher than that of OE (1.80 (0.3–325.4) vs. 0.80 (0.2–14.0), *p* < 0.001), and in the multivariate analysis, CEA was extracted as an independent factor to discriminate EAOC from OE, with the best cut-off as 1.25 ng/mL.

Recently, the Risk of Ovarian Malignancy Algorithm (ROMA) incorporated CA125 and HE4 with menopausal status in order to discern malignant from benign pelvic masses. Many studies have demonstrated the clinical utility of ROMA in women with a pelvic mass and have provided good accuracy in the risk stratification of epithelial ovarian carcinoma [28,29,30,31,32,33,34,35,36]. The ROMA showed more sensitivity in predicting malignancy in patients with high-grade serous carcinoma than endometrioid carcinoma and CCC histology [32,37]. In serous types of carcinoma, ROMA has good predictive efficacy (sensitivity = 75.0%, specificity = 89.3%), but in endometriotic types its efficacy is inferior (sensitivity = 61.9%, specificity = 75.3%) [38]. Our study showed comparable predictive efficacy with ROMA (sensitivity = 83.2%, specificity = 76.4%), and in multivariate analysis, the R2 predictive index incorporating CEA and tumor diameters revealed independent predictive factors in discriminating OE from EAOC.

This study has some limitations. The first limitation is that the R2 predictive index can only discriminate malignant endometriosis-associated ovarian tumors from benign ovarian endometriosis. Since borderline ovarian tumors or ovarian endometriosis with atypia cases are limited, more case accumulation is needed to assess the R2 predictive index. The second limitation is the inability to compare the efficacy of the ROMA and the R2 predictive index in the current study because HE4 has only recently been available in Japan. The third limitation is the retrospective nature of this study. To overcome these limitations, a prospective multi-institutional study is warranted.

## 6. Conclusions

In conclusion, the R2 predictive index is useful and valuable to discover the malignant transformation of endometrioma, which may be useful for all clinicians.

## Figures and Tables

**Figure 1 cancers-13-03829-f001:**
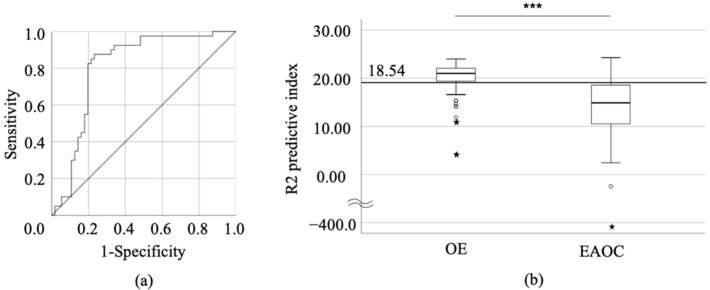
Analysis of correlation between pre-surgical factors and R2 value. ROC curve shows diagnostic effectiveness of R2 predictive index (**a**). (**b**) shows the distribution of R2 predictive index for each studied group as OE (*n* = 54) and EAOC patients (*n* = 51) (*p* < 0.001). *** *p* < 0.001 vs. control. The circles and pentagrams represent outliers.

**Figure 2 cancers-13-03829-f002:**
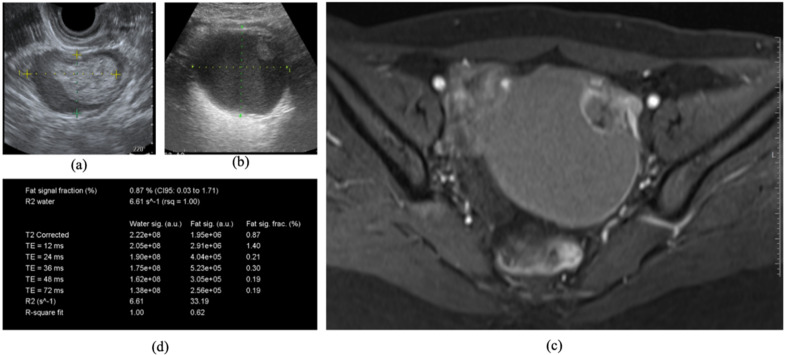
The patient underwent a routine ultrasound and MRI using T1W and T2W sequences. An MRI examination of the pelvis on a 3.0-Tesla MR unit (MAGNETOM Verio, Siemens Healthcare, Erlangen, Germany), using a pelvic-phased array coil followed. The mural nodules were demonstrated as polypoid structures. A cystic mass with hematoma (47 × 31 mm) was observed by ultrasound (**a**), but two years after the first administration, ultrasonography showed an enlarged ovarian cystic tumor (96 × 79 mm) with an solid portion (**b**). MRI identified the solid tissue components, detected with enhancing on dynamic contrast-enhanced images (**c**), and MR relaxometry demonstrated an R2 value of 6.61 s-1 in the cyst (**d**).

**Table 1 cancers-13-03829-t001:** Demographic and clinical characteristics of the investigative cohort.

	Benign Tumor	Malignant Tumor	*p*-Value
Number	*n* = 105	*n* = 37	
Age (years)			
Median (range)	38.00 (20–63)	48.00 (31–74)	
Mean ± SD	37.17 ± 9.31	49.62 ± 10.68	<0.001
BMI			
Median (range)	21.41 (15.01–30.46)	21.42 (17.01–31.22)	
Mean ± SD	21.51 ± 3.22	22.11 ± 3.78	0.528
Parity			
0	56	15	
≥ 1	49	22	0.126
FIGO sage	-	I (*n* = 29), II (*n* = 5), III (*n* = 2), IV (*n* = 1)	
Subtype	Endometriosis (*n* = 95)	With atypia cell (*n* = 3)	
	Dermoid (*n* = 5)	SMBT (*n* = 2)	
	Mucinous (*n* = 4)	CCC (*n* = 15)	
	Other (*n* = 1)	Endometrioid carcinoma (*n* = 10)	
		Mucinous carcinoma (*n* = 2)	
		Serous carcinoma (*n* = 3)	
		Seromucinous carcinoma (*n* = 2)	
Cyst size (mm)			
Median (range)	65.00 (21.00–193.00)	102.00 (38.99–231.92)	
Mean ± SD	65.63 ± 28.60	113.47 ± 51.79	<0.001
Menopause			
Yes	6	14	
No	99	23	<0.001

OE ovarian endometrioma, EAOC endometriosis-associated ovarian cancer, BMI body mass index, FIGO The International Federation of Gynecology and Obstetrics, SMBT seromucinous borderline tumor, CCC clear cell carcinoma, Hb hemoglobin.

**Table 2 cancers-13-03829-t002:** Demographic and clinical characteristics of the validation cohort.

	Benign Tumor (OE)	Malignant Tumor (EAOC)	*p*-Value
Number	*n* = 54	*n* = 51	
Age (years)			
Median (range)	38.00 (22–51)	54.00 (30–74)	
Mean ± SD	36.72 ± 7.45	54.24 ± 10.80	<0.001
BMI			
Median (range)	20.60 (15.61–35.75)	22.03 (15.37–32.40)	
Mean ± SD	21.62 ± 4.17	22.63 ± 3.83	0.092
Parity			
0	32	20	
≥ 1	21	31	0.031
FIGO sage	-	I (*n* = 32), II (*n* = 7), III (*n* = 10), IV (*n* = 2)	
Subtype	Endometriosis	With atypia cell (*n* = 1)	
		SMBT (*n* = 3)	
		CCC (*n* = 31)	
		Endometrioid carcinoma (*n* = 14)	
		Seromucinous carcinoma (*n* = 2)	
Cyst size (mm)			
Median (range)	68.00 (39.00–160.00)	111.00 (23.10–271.36)	
Mean ± SD	72.19 ± 23.43	111.09 ± 57.20	<0.001
Menopause			
Yes	0	34	
No	54	17	<0.001

OE ovarian endometrioma, EAOC endometriosis-associated ovarian cancer, BMI body mass index, FIGO The International Federation of Gynecology and Obstetrics, SMBT seromucinous borderline tumor, CCC clear cell carcinoma, Hb hemoglobin.

**Table 3 cancers-13-03829-t003:** Cut-off values discriminating EAOC from benign OE in the validation cohort.

	Cut-Off Value	Sensitivity	Specificity	PPV	NPV	AUC	*p*-Value
Age (years)	42	78.4	64.8	88.2	74.1	0.811	<0.001
Cyst size (mm)	85.00	67.6	84.8	66.7	79.6	0.796	<0.001
CEA (ng/mL)	1.3	57.1	66.2	60.4	77.5	0.665	0.006
Lymphocyte (/µL)	1391	73.4	66.7	62.0	64.8	0.723	<0.001
Hb (g/dL)	11.2	89.0	33.3	29.4	85.2	0.616	0.040
Platelet (×10^4^/µL)	29.3	52.8	79.0	39.2	79.6	0.672	0.002
CRP (mg/dL)	0.14	50.0	23.5	68.6	81.5	0.666	0.003
Alb (g/dL)	4.2	80.9	47.2	60.8	75.9	0.659	0.005
D-dimer (µg/mL)	0.8	64.7	76.7	76.5	57.4	0.709	<0.001
APTT (s)	28.0	50.0	75.0	56.9	64.8	0.641	0.014
R2 predictive index	18.54	82.4	68.6	75.0	85.0	0.813	<0.001

PPV positive predictive value, NPV negative predictive value, AUC area under curve, CEA carcinoembryonic antigen, CRP C-reactive protein, Hb hemoglobin, Alb albumin.

**Table 4 cancers-13-03829-t004:** Univariate and Multivariable analysis of the predictive factors of EAOC in the validation cohort.

		Univariate Analysis		Multivariate Analysis			
		Risk ratio (95% CI)	*p*-Value	Risk Ratio (95% CI)	*p*-Value	Risk Ratio (95% CI)	*p*-Value
Age (years)	<42	1.00 (referent)		1.00 (referent)		1.00 (referent)	
	≥42	21.43 (7.52–61.05)	<0.001	22.15 (5.02–97.6)	<0.001	17.20 (3.84–77.16)	<0.001
Cyst size (mm)	<85.00	1.00 (referent)				—	—
	≥85.00	7.82 (3.24–18.88)	<0.001			—	—
CEA (ng/mL)	<1.3	1.00 (referent)		1.00 (referent)		—	—
	≥1.3	5.26 (2.05–13.47)	0.001	4.49 (1.26–15.99)	0.020	—	—
Lymphocyte (/µL)	>1391	1.00 (referent)					
	≤1391	3.01 (1.35–6.68)	0.007				
Hb (g/dL)	>11.2	1.00 (referent)					
	≤11.2	2.40 (0.92–6.27)	0.075				
Platelet (×10^4^/µL)	<29.3	1.00 (referent)					
	≥29.3	2.52 (1.06–6.01)	0.037				
CRP (mg/dL)	<0.14	1.00 (referent)		1.00 (referent)		1.00 (referent)	
	≥0.14	9.63 (3.89–23.82)	<0.001	11.18 (2.80–44.64)	0.001	6.76 (1.58–28.89)	0.010
Alb (g/dL)	>4.2	1.00 (referent)					
	≤4.2	4.89 (2.11–11.32)	<0.001				
D-dimer (µg/mL)	<0.8	1.00 (referent)					
	≥0.8	4.38 (1.89–10.17)	0.001				
APTT (s)	>28.0	1.00 (referent)					
	≤28.0	2.43 (1.11–5.33)	0.027				
R2 predictive index	>18.54	1.00 (referent)		—	—	1.00 (referent)	
	≤18.54	17.00 (5.74–50.38)	<0.001	—	—	8.25 (2.13–32.02)	0.002

CEA carcinoembryonic antigen, Hb hemoglobin, CRP C-reactive protein, Alb albumin.

## Data Availability

The data presented in this study are available on request from the corresponding author.

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
