# Peer review of "A Novel Predictive Tool for Discriminating Endometriosis Associated Ovarian Cancer from Ovarian Endometrioma: The R2 Predictive Index"

_cancers, 2021, doi:10.3390/cancers13153829_

Round 1
Reviewer 1 Report
Among the main remarks to be made: the validation cohort presents a too high number of tumors compared to the investigative cohort, the authors should explain how they selected the cases of this cohort and explain why the number of tumors is so high. Among the minor observations that should be made: 1) it is necessary to make equal the rappresentative tables of the two cohorts, using benign tumor and malignant tumor in the first and OE and EAOC in the second is a confusing factor. 2) They do not specify the sensitivity and specificity of the investigational cohort, I assume it is in line with the literature as described in the introduction. 3) In the Patients section of the results they do not specify numbers and percentages of correctly diagnosed benign and malignant cases as in the investigational cohort.
Author Response
Among the main remarks to be made: the validation cohort presents a too high number of tumors compared to the investigative cohort, the authors should explain how they selected the cases of this cohort and explain why the number of tumors is so high. Among the minor observations that should be made: 1) it is necessary to make equal the rappresentative tables of the two cohorts, using benign tumor and malignant tumor in the first and OE and EAOC in the second is a confusing factor. 2) They do not specify the sensitivity and specificity of the investigational cohort, I assume it is in line with the literature as described in the introduction. 3) In the Patients section of the results they do not specify numbers and percentages of correctly diagnosed benign and malignant cases as in the investigational cohort.
Dear reviewer,
Thank you very much for your detailed and kind review.
I felt very appreciated by your comments, and please see below about the remarks.
It was very suggestive and improved remarkably our manuscript.
About the main remark, the reason why the malignant tumor cases are high in the validation cohort. I think that the number of malignant cases in the investigative cohort is relatively small, firstly because there were many cases where consent to conduct MR relaxometry were not obtained in malignant cases and secondly, the date when MR relaxometry can be conducted is restricted and the malignant cases conducted MR relaxometry was limited.
About minor remarks, I changed all of the suggested points.
Kind regards,
Reviewer 2 Report
The study is very interesting and has the potential to add value to the diagnostic of ovarian cancer. However, there a major limitations in terms of usefulness and the paper should benefit from English editing. First, perhaps the R2 index score could be tested/used complementary to the MRI/US and not alone as a predictive tool, otherwise, how can we select only the endometriosis patients that could benefit from it? Second, the index is based on the tumour diameter, measured on MRI. Without using MRI (because this is what the authors aim), I assume that the tumour diameter should be measured on US (endovaginal, abdominal?), and tested again to see if the R2 index maintains its specificity and sensitivity.
- Line 10 -> It is well known that contrast-enhanced MRI (especially subtraction images derived from unenhanced and contrast-enhanced T1-weighted imaging) is used to detect the cancer that is developing within an endometrioma. MR Relaxometry, is still a matter of debate, an experimental method that still needs proper validation. The author’s affirmation should be based on strong evidence, rather than a few papers (2), including one case report. However, enhancing mural nodule within an endometrioma are sensitive, but not very specific. Different pitfalls (eg. decidual reaction of pregnancy) should be mentioned. Here is the point where the R2 index could add value to the diagnosis.
- Line 67 -> This is a retrospective study, for which the authors claimed that a “written informed consent was obtained from all 247 patients.” –> I found this questionable.
- Line 86 - “After the routine clinical MR imaging, the registered patients underwent MR relaxometry” -> The routine MRI had contrast-enhanced sequences? If so, the authors should mention it. How about the interference of the contrast-enhancement agents with the relaxometry times?
- In the first cohort (142 patients), the post-operative diagnosis of OE was found in 95 (90.5%) and EAOC in 32 (91.4%) patients -> I think it is a mistake here.
- Line 100 – The imaging protocol is not clear enough. The VOI spectroscopy voxel, was placed on the relaxometry sequences? What about cystic lesion that were less than 15mm, because endometriomas are sometimes very small.
- Line 140 – the authors should mention how did obtain the equations of R2 values related to pre-surgical factors. Also, in the multivariate analysis, CRP and Albumin obtained statistical p-value. Why the predictive score did not include them?
- The case report should be a separate part, and not included into original, research study. If I understand correctly, a young patient with a suspicious ovarian tumour was followed-up for two years, because the R2 index suggested a benign ovarian endometrioma? No MRI at that point? However, the patient informed consent about the MR Relaxometry should be addressed, especially after the diagnostic MRI: “MRI identified the solid tissue components, detected with enhancing on dynamic contrast-enhanced images… The patient was 195 advised to have MR relaxometry of the pelvis”
- Line 259 - “Our study showed greater predictive efficacy than ROMA (sensitivity 259 = 83.2%, specificity = 76.4” -> But what was the ROMA score in your cohort of patients? Perhaps your patients larger tumours (>7cm) could represent a bias in your study.
- Line 263 - The first limitation questions the added value of the whole study. The authors aim to develop a score that is able to detect endometriosis-associated cancer, without imaging. But how can the score be used, if it is only feasible for documented ovarian endometrioma?
- The article should benefit from English editing. Sometimes the phrases are difficult to understand: “Furthermore, in the second cohort, no delivery history was more in OE patients”/ “CEA was extracted as an independent factor to discriminate EAOC from OE 249 with the best cut-off was 1.25 ng/ml.”
Author Response
The study is very interesting and has the potential to add value to the diagnostic of ovarian cancer. However, there a major limitations in terms of usefulness and the paper should benefit from English editing. First, perhaps the R2 index score could be tested/used complementary to the MRI/US and not alone as a predictive tool, otherwise, how can we select only the endometriosis patients that could benefit from it? Second, the index is based on the tumour diameter, measured on MRI. Without using MRI (because this is what the authors aim), I assume that the tumour diameter should be measured on US (endovaginal, abdominal?), and tested again to see if the R2 index maintains its specificity and sensitivity.
Dear reviewer,
Thank you very much for your detailed and kind review.
I felt very appreciated by your comments, and I followed almost all of your suggestions.
It was very suggestive and improved remarkably our manuscript, and I’m willing to undergo the publisher’s English editing.
Please see the below about the first concern. And for the second concern, I found that some tumor diameters were not calculated on ultrasound images because there were already prepared MRI results at first administration to our institution. There will be a bias to estimate the diameter using a reference line on an ultrasound image.
Kind regards,
- Line 10 -> It is well known that contrast-enhanced MRI (especially subtraction images derived from unenhanced and contrast-enhanced T1-weighted imaging) is used to detect the cancer that is developing within an endometrioma. MR Relaxometry, is still a matter of debate, an experimental method that still needs proper validation. The author’s affirmation should be based on strong evidence, rather than a few papers (2), including one case report. However, enhancing mural nodule within an endometrioma are sensitive, but not very specific. Different pitfalls (eg. decidual reaction of pregnancy) should be mentioned. Here is the point where the R2 index could add value to the diagnosis.
You are right. I changed the sentence more clearly to appeal to the usefulness of the R2 index.
- Line 67 -> This is a retrospective study, for which the authors claimed that a “written informed consent was obtained from all 247 patients.” –> I found this questionable.
The consent form of patients’ data availability for research use is obtained at the first administration, and after approval of this study by the Ethics Review Committee, the opt-out form was conducted through our institutional homepage.
- Line 86 - “After the routine clinical MR imaging, the registered patients underwent MR relaxometry” -> The routine MRI had contrast-enhanced sequences? If so, the authors should mention it. How about the interference of the contrast-enhancement agents with the relaxometry times?
This method is plane MR imaging and MR relaxometry also does not need enhancing agents.
- In the first cohort (142 patients), the post-operative diagnosis of OE was found in 95 (90.5%) and EAOC in 32 (91.4%) patients -> I think it is a mistake here.
Yes. I fixed the sentence.
- Line 100 – The imaging protocol is not clear enough. The VOI spectroscopy voxel, was placed on the relaxometry sequences? What about cystic lesion that were less than 15mm, because endometriomas are sometimes very small.
Yes. I understand your concern and modified it more clearly.
- Line 140 – the authors should mention how did obtain the equations of R2 values related to pre-surgical factors. Also, in the multivariate analysis, CRP and Albumin obtained statistical p-value. Why the predictive score did not include them?
The regression analysis was conducted to make a model formula predicting the R2 value by each factor. By multiple regression analysis, only the tumor diameter and CEA was extracted to make the predictive formula, and CRP and albumin were not extracted.
- The case report should be a separate part, and not included into original, research study. If I understand correctly, a young patient with a suspicious ovarian tumour was followed-up for two years, because the R2 index suggested a benign ovarian endometrioma? No MRI at that point? However, the patient informed consent about the MR Relaxometry should be addressed, especially after the diagnostic MRI: “MRI identified the solid tissue components, detected with enhancing on dynamic contrast-enhanced images… The patient was 195 advised to have MR relaxometry of the pelvis”
I presented the case report to show the effectiveness of the predictive R2 value. The case had been observed for two years as a benign ovarian endometrioma, and the first time MRI conducted was at the point of emerging the solid part. This case shows that the predictive R2 value could be used observation over time. I inserted the sentence into the suggested part.
- Line 259 - “Our study showed greater predictive efficacy than ROMA (sensitivity 259 = 83.2%, specificity = 76.4” -> But what was the ROMA score in your cohort of patients? Perhaps your patients larger tumours (>7cm) could represent a bias in your study.
In Japan, because HE4 had recently been made available, the ROMA index in our study can not be calculated. I compared just the numerical values. This description is not correct, so I changed it.
There was large benign ovarian endometrioma and confirmed by pathological examination.
- Line 263 - The first limitation questions the added value of the whole study. The authors aim to develop a score that is able to detect endometriosis-associated cancer, without imaging. But how can the score be used, if it is only feasible for documented ovarian endometrioma?
Clinicians can diagnose ovarian tumors as endometrioma just by ultrasound device because it’s characteristic imaging. And for the observation period or LEP/DNG intervention period this score can be useful to decide the point of surgical intervention as a landmark.
- The article should benefit from English editing. Sometimes the phrases are difficult to understand: “Furthermore, in the second cohort, no delivery history was more in OE patients”/ “CEA was extracted as an independent factor to discriminate EAOC from OE 249 with the best cut-off was 1.25 ng/ml.”
I understand your concern. I would like to undergo the English editing service of the publisher.

Round 2
Reviewer 1 Report
Now the paper is suitable for pubblication according to me.
Reviewer 2 Report
In its present format, the manuscript is not suitable for publication. I recommend significant revisions or new data in the manuscript to warrant further consideration for publication of this manuscript.